# Preparation and Characterization of a Composite Dust Suppressant for Coal Mines

**DOI:** 10.3390/polym12122942

**Published:** 2020-12-09

**Authors:** Hu Jin, Yansong Zhang, Kun Chen, Kuo Niu, Guangan Wu, Xiangrui Wei, Houwang Wang

**Affiliations:** 1College of Energy and Mining Engineering, Shandong University of Science and Technology, Qingdao 266590, China; 15764250719@163.com; 2College of Safety and Environmental Engineering, Shandong University of Science and Technology, Qingdao 266590, China; a990448700@163.com (K.C.); nksdust@163.com (K.N.); guanganwu@163.com (G.W.); 15684068939@163.com (X.W.); whw11052512@163.com (H.W.); 3Qingdao Intelligent Control Engineering Center for Production Safety Fire Accident, Qingdao 266590, China

**Keywords:** surfactant, soy protein isolate, chemical modification, composite dust suppressor, characterization of properties

## Abstract

In an effort to effectively control coal dust pollution and thereby reduce the harm of coal dust to human health, we prepared a highly efficient composite dust suppressant. First, dynamic contact angle and zeta potential measurements were used to select sodium dodecyl sulfonate (SDS) over sodium carboxymethyl cellulose and trisodium methyl silicon as the complementary additive to soy protein isolate for the dust suppressant. We employed viscosity and wind erosion resistance tests to compare the performance of the composite dust suppressant with three common, commercially available suppressants. As the concentration of the composite dust suppressant was increased, the viscosity increased, reaching a maximum value of 22.7 mPa·s at a concentration of 5 wt%. The 5 wt% concentration of the composite dust suppressant provided the lowest wind erosion rate (20.62%) at a wind speed of 12 m/s. The composite dust suppressant also had good bonding performance and wind erosion resistance. Scanning electron microscopy, X-ray diffraction, and thermogravimetric analysis were used to characterize the properties of the dust suppressants. The dust suppressant, which had a crystal-like structure, could easily capture coal dust and form an effective package. In addition, the density of the dust suppressant film increased as its crystallinity increased. The increased density was beneficial in that it enabled the dust suppressant to form a hard, solidified shell on the surface of coal dust, which improved dust suppression. The composite dust suppressant also had good thermal stability.

## 1. Introduction

A large amount of dust is generated during the production, transportation, and storage of coal. The dust seriously reduces the safety and efficiency of coal mines [1,2,3,4,5]. In recent years, as both the mining depth and the degree of mechanization in coal mining has increased, the generation and transmission of underground coal dust has become more complicated. Coal dust seriously affects human health, causes coal dust explosion accidents, and reduces the service lifetime of mechanical equipment. Additionally, open-air coal storage yards can be blown by strong winds to form an open source of pollution and dust, causing serious environmental pollution [6,7,8,9]. In particular, respirable dust, with a particle size of less than 5 μm, has a high degree of dispersion. Respirable dust has a large specific surface area and a strong adsorption capacity, and it can remain airborne for a long period of time. Operator exposure in this environment for a long period of time can cause pneumoconiosis [10,11]. At the same time, with the global outbreak of 2019-nCoV, 2019-nCoV virus can adhere to the dust to form aerosol, which can be suspended in the air and easily be inhaled by human body and cause infection, as shown in Figure 1 [12,13]. Therefore, the dust hazard is an urgent problem to be solved.

Water spraying is often used to reduce dust in both underground and open-air environments in coal mines. However, because coal is hydrophobic, the effect of spraying water is limited in terms of dust suppression. With the continuous development of dust suppression technology, chemical dust suppression methods have attracted more attention and have become effective, economical, and practical [14,15,16,17] Han et al. optimized three types of surfactants, and explored the influence of the combination of surfactants on coal dust suppression through different compounding methods [18]. Jiang et al. formulated a foam dust suppressant with coconut oil monoethanolamide, sodium α-olefin sulfonate, fatty alcohol polyoxyethylene ether sodium sulfate, and lauryl dimethyl betaine. Their dust suppressant could effectively inhibit dust in open air that was generated during down-the-hole drilling of a mine, with an average dust removal rate of 87.8% [19]. Yang et al. formulated a polymer dust suppressant with 0.1% Triton X-100 and 0.7% guar gum that was suitable for brown coal open-pit mines. The dust suppressant had an obvious effect on the suppression of fine particles, and the hardness of its cured layer reaches 61.43, which can resist winds at 7 on Beaufort wind scale [20]. Krzysztof et al. studied the effects of anionic, cationic, and nonionic surfactants on the charged dust removal performance of different coal powders and different coal particle sizes [21]. Inyang et al. conducted experiments on montmorillonite and kaolin with sodium hydroxymethyl cellulose, polyacrylamide, and polyoxyethylene, and their studies revealed that different types of dust should use different types and concentrations of dust suppressants [22]. Magnusson et al. used lignosulfonate and chloride as raw materials to form a dust suppressant that had a clear dust suppression effect on gravel pavement [23].

Chemical dust suppression is a new method of controlling dust pollution. Dust suppression agents have been regarded as an important means to effectively solve the problem of open dust sources. Most dust suppressants have good dust suppression abilities, but also have problems such as cost, a single use nature, toxicity, lack of biodegradability, corrosiveness, and secondary pollution [24,25,26]. To overcome some of these problems, a high molecular weight polymer soy protein isolate was selected for this work. It is biodegradable, natural, and renewable. Through chemical modification, a high-efficiency and environmentally-friendly composite dust suppressant was developed, as shown in Figure 2. Compared with common dust suppressants on the market, this dust suppressant had good viscosity and wind erosion resistance. The dust suppressant particles, which have a crystal-like structure, have high crystallinity and good film density [27,28,29]. They can effectively capture and coat coal dust to form a hard, solidified shell that has good thermal stability. The composite dust suppressant developed by this research, referred to as CDS, is simple in preparation, environmentally friendly, and has excellent dust suppression effects. It follows the theme of the era of green environmental development.

## 2. Materials and Methods

### 2.1. Main Reagents and Equipment

The reagents and equipment used in the experiment are shown in Table 1 and Table 2.

### 2.2. Test of Coal Dust Particle Size

The samples were collected from coal dust near the coal storage yard of Zaozhuang Coal Mine in Shandong Province, Zaozhuang, China. In order to visually find out the particle size distribution of coal samples intuitively, all particle sizes of all coal dust samples were measured. In this paper, the laser particle size analyzer Mastersizer 3000 is used to analyze the particle size of coal dust.

### 2.3. Optimization of Surfactants

This research initially evaluated three different surfactants, each with good wetting properties: Polyoxyethylene ether (AEO), cetyltrimethylammonium bromide (CTAB), and sodium dodecyl sulfonate (SDS). Dynamic contact angle and zeta potential measurements were conducted to explore the wetting ability of the surfactants on coal dust. The surfactant with the best wetting ability was selected as the complementary reagent in the synthesis of the dust suppressant.

#### 2.3.1. Dynamic Contact Angle Test

Firstly, 550 mg pulverized coal was pressurized to 500 t/cm^2^ for 3 min to make coal cake. The diameter of coal cake was 13 mm, the thickness was 2~3 mm, and the surface was smooth. The coal cake was put on the slide of the sample table of the horizontal contact angle measuring instrument. The sample table was lifted slowly. When the sample plate was about to touch the liquid drop, the operating instrument started to take pictures.

#### 2.3.2. Zeta Potential Test of Coal Dust

(1)First, surfactant solutions of different concentrations were prepared in deionized water. Each equal amount of coal powder was put into the surfactant solution for ultrasonic dispersion for 20 min, and the mixture was configured into a uniform suspension. The suspension was prepared in a neutral environment at a room temperature of 23 °C.(2)The suspension was poured into the sample cell and the lid closed, and the sample cell was put into the zeta potential measuring instrument.(3)The test software was opened, the parameter values of the sample to be tested were set, such as shading, refractive index, concentration, dielectric constant, etc., and three measurements were taken.(4)After waiting for the result to come out, the data was recorded and the results saved.

### 2.4. Preparation of Composite Dust Suppressant

Preparation process: We added 7.5 g of soy protein isolate (SPI), 0.15 g of sodium carboxymethyl cellulose, 10 mL of 0.2 wt% sodium dodecyl sulfonate (Obtained by surfactant optimization experiment), and 140 mL of deionized water into a three-necked flask. The flask was fixed on the thermostatic magnetic stirrer and stirred for one hour in a constant temperature environment of 60 °C. After the solution was cooled, 2.5 g of 30 wt% sodium methyl silicate was added and stirred evenly to obtain the dust suppressant solution [30,31,32,33].

During the experimental reaction, the stirring speed was controlled to be 60–80 r/min to prevent the solution from foaming. The experimental flowchart is shown in Figure 3.

### 2.5. Viscosity Experiment Test of Dust Suppressant

In order to verify the dust suppression effect of the developed dust suppressant, the experiments selected the commonly-used crust-type dust suppressant (CTDS), mine road dust suppressant (MRDS), efficient environmental dust suppressant (EEDS), and the developed CDS. To compare and test the viscosity and wind erosion resistance respectively.

First of all, viscosity values of four dust suppressors were tested. The viscosity values of four dust suppressor solutions at the concentration of 1, 2, 3, 4 and 5 wt% were measured by NDJ-79 rotary viscometer, and the average value was taken after three tests. 

### 2.6. Anti-Wind Erosion Test of Dust Suppressant

The experiment used a dust-proof laboratory and a wind tunnel simulation platform to determine the on-site wind erosion rate, as shown in Figure 4. First, a petri dish was filled with a weighed amount of coal powder. A 5 wt% solution of dust suppressant in water was sprayed uniformly on the surface of the sample, and the sample was dried after a period of reaction. The wind speed was set to 3–12 m/s to simulate natural wind, and it blew on the sample continuously for 0.5 h. After the wind treatment the sample was weighed to calculate the wind erosion rate, according to the quality of the coal loss rate. The results are shown in Table 3. To analyze the anti-wind erosion effect of each dust suppressant, its experimental data was linearly fitted.

### 2.7. Characterization Method of Dust Suppressant

(1)Observation with scanning electron microscope (SEM)

We used an Apreo high-resolution SEM to observe the surface morphology of the sample. A CDS solution was sprayed evenly on the surface of coal dust, and allowed to dry for 12 h in ambient conditions before examination in the SEM. Magnifications of 5000, 15,000, and 30,000 were used to observe the structural characteristics and surface morphology of the coal samples. SEM was also used to examine the dust suppressant.

(2)X-ray diffraction test (XRD).

A Rigaku Utima IV XRD was used to analyze the crystallization of the CDS film. For XRD analysis, a uniform, flat film was fixed on the XRD metal sample plate. The XRD scanning range for 2θ was 10–60°. In the experiment, equal amounts of the dust suppressant and the soy protein isolate, at concentrations of 2 and 5 wt%, respectively, were sprayed in the petri dish. The sprayed material was dried, and the complete membrane was taken out for observation.

(3)Thermogravimetric test (TGA).

A LABSYS evo TGA was used to analyze the thermal stability of CDS. First, a TGA crucible was heated with an alcohol burner to remove impurities. A small sample of the dried composite was placed in the crucible, which was hung on the balance of the TGA system. The mass of the sample was 10.83 mg. The sample was heated in N_2_ gas at 10 °C/min from room temperature to a maximum temperature of 800 °C.

## 3. Experimental Results

### 3.1. Coal Dust Particle Size Analysis

In order to observe the particle size distribution of coal dust more intuitively, we performed data and curve fitting, as shown in Figure 5.

It can be seen from Figure 5 that the particle size distribution of the tested coal dust contains three primary peaks. The first peak appears in the range of 0.214–0.991 µm, and the second peak is in the range of 4.580–16.400 µm. The third, more distinct peak was in the range of 400–1110 µm. Dust with particle sizes of 0–5 µm is generally referred to as “respirable dust”, which causes greater harm to human health than larger dust particles. Respirable dust can reach the lungs through the upper respiratory tract of the human body, deposit in the alveoli, and accumulate for a long time to cause various lung diseases. By analyzing the experimental data of dust particle size, a large amount of respirable dust < 5 µm in size exists in the mining environment, causing environmental pollution in the mining area. Therefore, effective measures must be taken to control dust pollution to ensure the normal operation of coal mines and create good, healthy production environments. To address the problem of environmental pollution caused by airborne dust, we experimentally developed a new type of composite dust suppressant for mines.

### 3.2. Dynamic Contact Angle Experiment Analysis

The surface of coal dust is strongly hydrophobic because it contains many hydrophobic groups, such as aliphatic and aromatic hydrocarbons. Therefore, it is difficult for coal dust to be wetted by water. Surfactants can be added during the synthesis of a composite dust suppressant. The main function of the surfactants in a composite dust suppressant is to wet the coal dust and increase its hydrophilicity. Because surfactants have both hydrophobic and hydrophilic groups, a surfactant is able to interact with both the hydrophobic coal and the hydrophilic water. The hydrophobic groups on the surface of coal dust interact with the hydrophobic groups of the surfactant. Simultaneously, the hydrophilic moieties of the surfactants can interact with water. Thus, the combination of hydrophilic and hydrophobic properties of surfactants make it easier for a composite dust suppressant to wet the surface of coal dust [34,35,36]. The contact angle of aqueous solutions of the three types of surfactant at the same concentration, along with pure water as a control group, were measured on pre-fabricated briquettes. The test results are shown in Figure 6.

As shown in Figure 6, the contact angle of pure water was the largest at 75.91°. This indicates that the wetting effect of clear water was the worst, and it was difficult for water to permeate the coal dust. The anionic surfactant SDS had the smallest contact angle of 40.09°, indicating that it had a good wetting ability on coal dust and it was easy for it to permeate the coal dust.

### 3.3. Zeta Potential Test Analysis

In order to further optimize the surfactants with good wetting effect on coal dust, the influence of three surfactants on the zeta potential of coal dust was studied, and the optimal dust suppressant additives were determined by comprehensive test results. The zeta potential experiment uses the stern model. The diffusion double layer can be divided into two layers, one is the dense layer close to the particle surface, also called the Stern Layer; the other layer is called the diffuse layer [37,38]. as shown in Figure 7.

The calculation formula of potential value is:(1)$$U_{E}=\frac{2\varepsilon \zeta }{3\eta }g(\kappa a)$$

In Equation (1), the dielectric constant value is 80, the viscosity value is 2.6 mP·s, the κa value is 1.5, and the temperature is set to 26 °C. Surfactants were added to the experimental water to different concentrations (0, 0.1%, 0.2%, 0.3%, 0.4%, and 0.5%), and then moved into the sample pool to prepare for measurement, using a Malvern particle size potentiometer to measure three times to get the average value. The experimental results are shown in Figure 8.

It can be seen from Figure 8 that the three different surfactants had different effects on the zeta potential of coal dust. Among them, the non-ionic surfactant, AEO, had little effect on the zeta potential of a pulverized coal solution. The cationic surfactant, CTAB, increased the zeta potential in the pulverized coal solution, and the potential value at point A gradually changed from negative to positive. Meanwhile, as the solution concentration continued to increase, the increase in the potential value gradually slowed after the concentration reached 0.2%, (slope point B in Figure 8). The anionic surfactant, SDS, reduced the zeta potential in the pulverized coal solution, and the change in the zeta potential of the coal dust solution gradually slowed when the concentration reached 0.2% (slope point C in Figure 8).

The zeta potential analysis revealed that the nonionic surfactant solution exhibited electrical neutrality. The nonionic surfactant had little effect on the zeta potential of coal dust. For the cationic surfactant CTAB, the hydrophilic group had a positive charge, and the adsorption of the surfactant on the surface of coal dust particles mainly depended on static electricity. The positively charged hydrophilic group was adsorbed on the surface of the coal particles, while the hydrophobic group faced outward. The positive charge carried by its ions could neutralize the negative charge on the surface of the coal powder, thus reducing and to some extent reversing the negative electric charge of the coal dust in the solution. In contrast, In the anionic surfactant solution, the anionic surfactant will dissociate negatively charged surface active ions in the water. They are adsorbed on the surface of coal dust with their tails facing the surface of coal dust and their heads pointing to the solution, making the surface of coal dust negatively charged stronger, as shown in Figure 9. This extension of the hydrophilic group into the water phases increased the wetting ability of SDS, and the negative charge of the ions increased the electronegativity of the coal powder in the solution [39,40,41,42].

Analysis of the test data showed that among the three surfactants, SDS could increase the electronegativity of coal dust. The surfactant was firmly adsorbed on the surface of coal dust to promote the wetting of the coal dust. Based on the test results of the contact angle and zeta potential measurements, a 0.2 wt% concentration of SDS was selected as the complementary additive for the developed composite dust suppressant, CDS, used in subsequent experiments.

### 3.4. Viscosity Experiment Analysis

The statistical results are shown in Figure 10, according to the analysis in Figure 10, the viscosity of CDS solution at a concentration of 1% was less than that of the commercially available crust-type dust suppressants. However, as the concentration was increased, the viscosity of the CDS solution exceeded that of the other three dust suppressants. The CDS solution viscosity reached a maximum of 22.7 mPa·s at concentration of 5 wt%. The viscosity results show that when the concentration of CDS exceeds 1 wt%, it has good bonding properties. Compared with three commercially available dust suppressant solutions, the molecules in the solution have stronger interaction [43,44,45,46,47].

### 3.5. Analysis of Anti-Wind Erosion Test of Dust Suppressant

To further evaluate the dust suppression capability of CDS, anti-wind erosion performance tests were conducted. The anti-wind erosion test is an important index to test the performance of dust suppressants. The wind erosion resistance of dust suppressants can be expressed by measuring the wind erosion rate, which has significance as a reference value for the research and development of dust suppressants in practical applications.

As shown in Table 3 and Figure 11, as the wind speed increased, the coal loss rate of the four types of dust suppressants increased linearly. The newly developed composite dust suppressant had the highest degree of fit with R^2^ = 0.9966. In general, the wind erosion rate increased linearly with the increase of wind speed. However, the wind erosion rate of the composite dust suppressant was lower at the same wind speed than the wind erosion rate of the commercially available dust suppressants. For example, when the wind speed was 12 m/s, the wind erosion rate varied from small to large: CDS(20.62%) < CTDS(22.87%) < MRDS(23.46%) < EEDS(29.69%). These results show that CDS can provide better dust suppression than the three commercially available dust suppressants when the wind speed is high.

Through the measurement of viscosity and anti-wind erosion, the performance and dust suppression effect of the four tested dust suppressants can be comprehensively compared. CDS had the highest viscosity and the best anti-wind erosion effect, which can effectively inhibit dust pollution.

### 3.6. Characterization Results and Discussion

(1)The morphology of the surface of coal dust sprayed with CDS as well as the surface of CDS was observed with SEM. The scanned image is shown in Figure 12.

It can be seen from Figure 12 that (a), (b), and (c) are morphological images of 5000, 15,000, and 30,000 respectively enlarged by the dust suppressant sprayed on the surface of coal dust. It can be seen from positions 1 and 2 that CDS tightly binds small and large coal dust particles together, showing a strong bonding force. In position 3, it can be seen that CDS covers the surface of coal dust uniformly and forms a dense, solidified shell on the surface of coal dust, which enables CDS to resist wind erosion and achieve good dust reduction. The (d) figure is an SEM image of dry CDS. From position 4 it can be seen that the structure of CDS presents a relatively regular flocculant shape, similar to a crystal structure. This structure can increase the dust catching ability. When the dust comes into close contact with CDS, CDS can effectively bond to the dust and cover it. CDS eventually inhibits dust pollution by permeating the dust particles, bonding them together, and consolidating them into a shell.

(2)The crystallization of soy protein isolate and CDS after film formation was observed with XRD.

As shown in Figure 13, the XRD curves of three samples of SPI membrane, 2 wt% dust suppressor solution and 5 wt% dust suppressor solution have a broad main peak at 2θ = 20° nearly. There were no obvious new diffraction peaks in other regions. The A diffraction peak of the soy protein isolate film was relatively flat, indicating that the film was a polymer with low crystallinity. The B and C diffraction peaks of CDS obtained after modification of the soybean protein isolate gradually became narrower and sharper than the A diffraction peak. Also, as the concentration of CDS increased, the C diffraction peak became narrower than the B diffraction peak, indicating that the crystallinity of the CDS film from the 5 wt% solution was higher than that from the 2 wt% solution. This finding suggests that as the concentration of CDS increases, its crystallinity increases.

The analysis shows that the soy protein isolate is a partially crystalline material with relatively low crystallinity. Modification of the soy protein isolate changes the spatial structure of the peptide chain, which can open the agglomerated and folded soy protein isolate peptides and reduce the cohesion of the soy protein isolate. However, the soy protein isolate peptide chains can be fully stretched and orderly arranged, and therefore it is easier to crystallize. The increase in film crystallinity will enhance the density of the film, which will help the dust suppressant form a hard shell on the surface of the coal dust to improve dust reduction.

(3)The TG-DTG curve is obtained by testing the dust suppressant samples. As shown in Figure 14.

As can be seen from Figure 14, the thermal decomposition process of CDS was roughly divided into four stages: Dehydration, slight weight loss, rapid weight loss, and carbonization. CDS was in the dehydration stage when the temperature was 30–143 ℃, and the weight loss was 9.73%. As the temperature increased, the rate of weight loss first increased and then decreased, and an obvious dehydration peak appeared. The CDS sample reached a maximum dehydration rate at 114 ℃, which was mainly caused by the evaporation of free water and bound water in the sample. When the temperature was between 143–206 ℃, the pyrolysis was in the transitional stage, and the TGA curve tended to be stable. The most important pyrolysis weight loss stage was between 206 and 546 ℃. The rate of weight loss at this stage accounted for 49.4% of the total weight loss of pyrolysis. In the carbonization stage, as the temperature increased, the TGA curve of CDS tended to be flat, and the pyrolysis rate was close to zero on the difference thermogravimetry curve. The TGA results show that CDS has good thermal stability, such that it can meet daily production needs at temperatures ranging from room temperature to 100 ℃.

## 4. Conclusions

(1)Dynamic contact angle measurements showed that the anionic surfactant SDS had good wetting of coal briquette samples, with a minimum contact angle of 40.09°. SDS reduced the zeta potential of a pulverized coal solution, increasing its electronegativity and also its wetting ability. Among the three surfactants tested, the anionic surfactant SDS at a concentration of 2 wt% was preferred as an additive to the for the composite dust suppressant CDS.(2)The performance of CDS prepared by the chemical modification method was compared to common, commercially available chemical dust suppressants. It was found that the viscosity of the four different types of dust suppressants increased with an increase in dust suppressant concentration. CDS had a maximum viscosity of 22.7 mPa·s at a concentration of 5 wt%. Moreover, it had relatively good anti-wind erosion ability, good dust suppression capacity, and could effectively inhibit dust pollution.(3)SEM, XRD, and TGA were used to characterize the properties of CDS. Modification of the soy protein isolate caused its crystallinity density to improve. After spraying the coal dust surface with CDS, it could effectively coat the coal dust to form a dense and hard solidified shell. CDS exhibited good wind erosion resistance and had good thermal stability, indicating that it should be adaptable to harsh environments.(4)The soy protein isolate used in the experiment is a green and environment-friendly polymer material with a wide range of sources, a relatively low processing cost, and a low price, and it is also biodegradable.

## Figures and Tables

**Figure 1 polymers-12-02942-f001:**
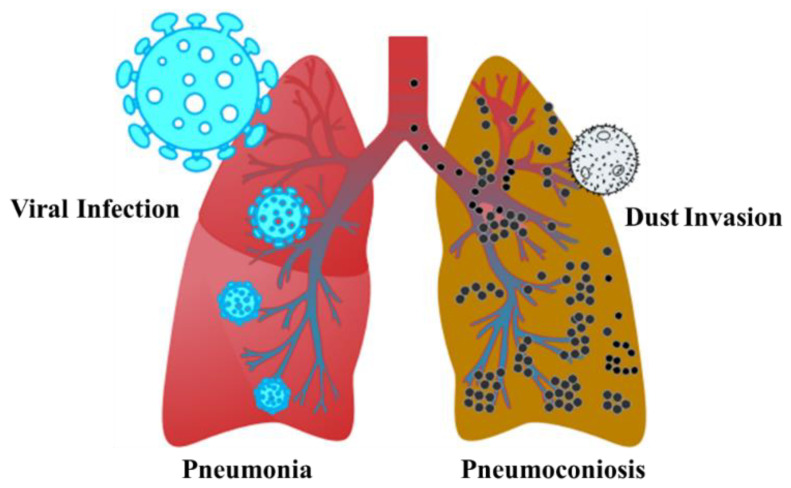
Diagram of pneumoconiosis and pneumonia.

**Figure 2 polymers-12-02942-f002:**
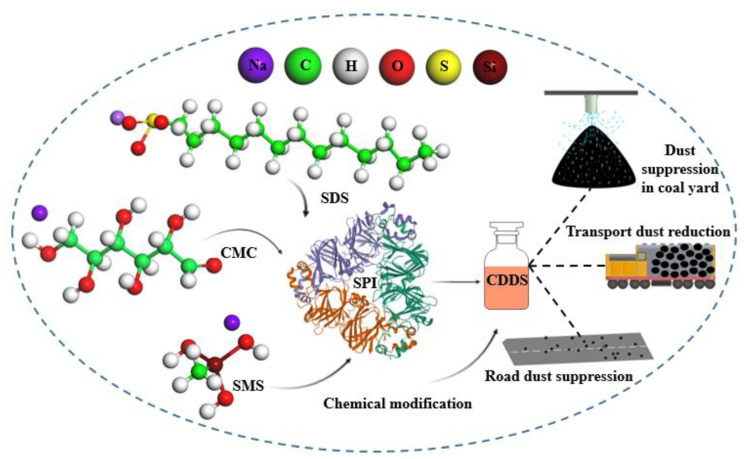
Schematic diagram of dust suppressor preparation.

**Figure 3 polymers-12-02942-f003:**
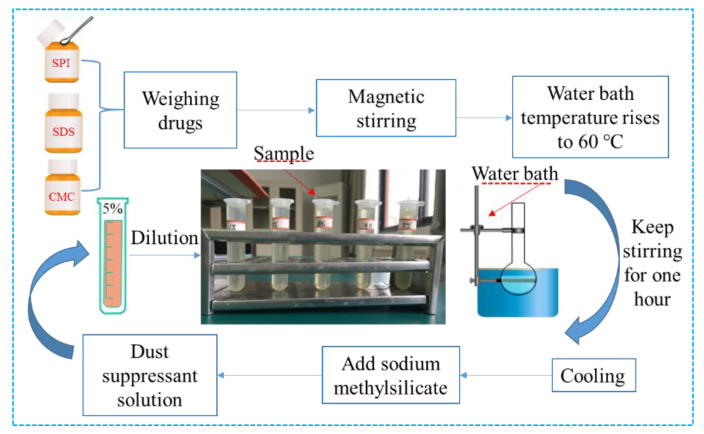
Flow chart of preparation of dust suppressant.

**Figure 4 polymers-12-02942-f004:**
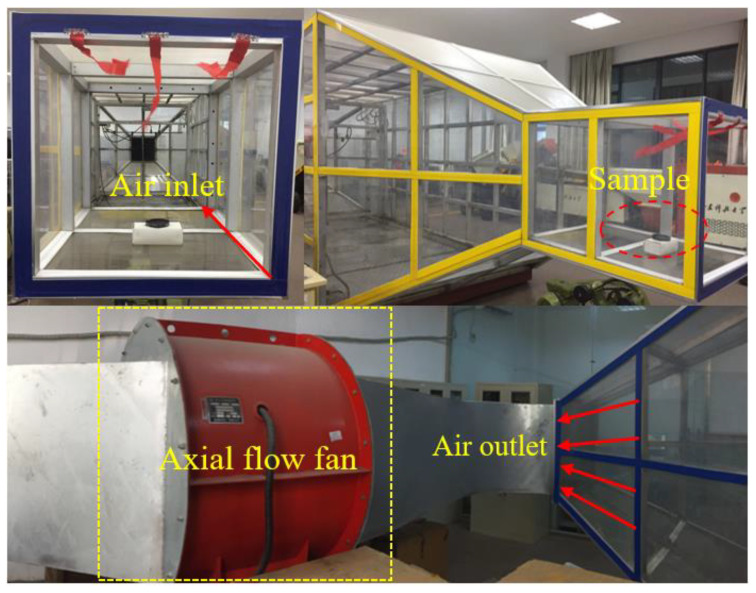
Layout drawing of the erosion rate testing experiment equipment.

**Figure 5 polymers-12-02942-f005:**
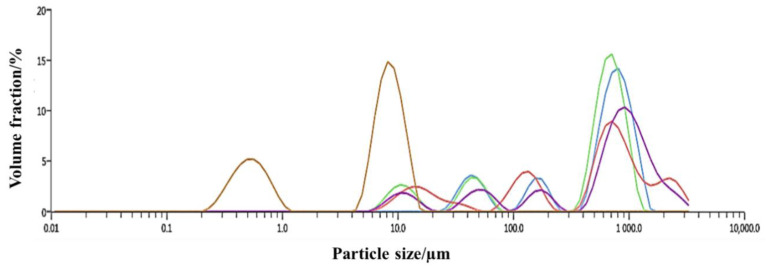
Trend chart of data curve.

**Figure 6 polymers-12-02942-f006:**
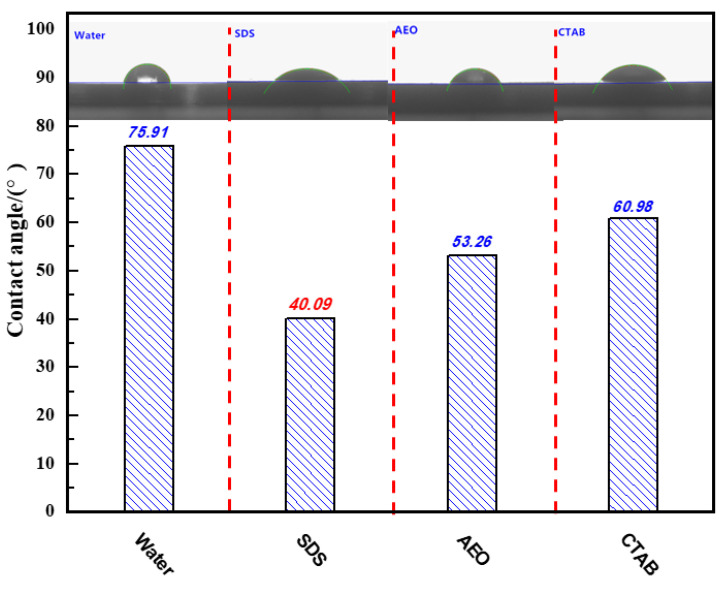
Contact angles of different surfactants on coal samples.

**Figure 7 polymers-12-02942-f007:**
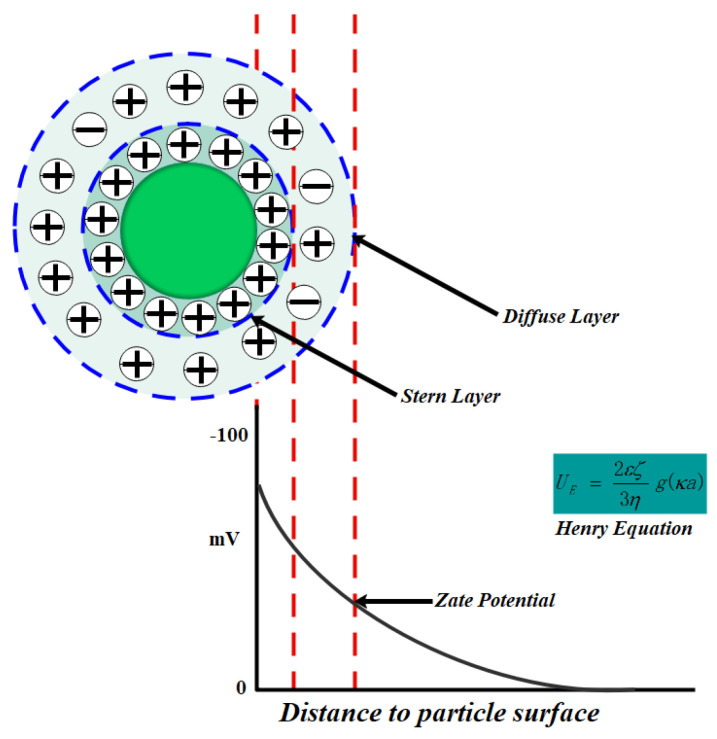
Stern model.

**Figure 8 polymers-12-02942-f008:**
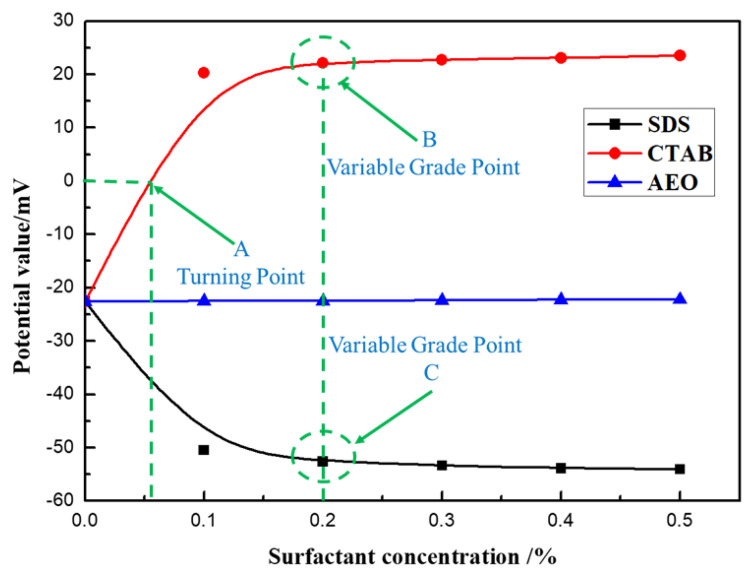
The zeta potential of coal dust under different surfactant concentrations.

**Figure 9 polymers-12-02942-f009:**
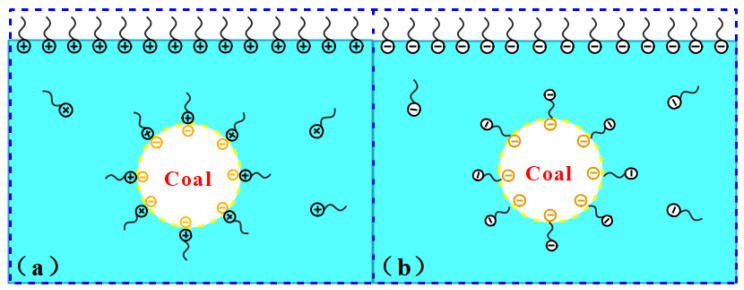
Schematic diagram of adsorption of surfactant on coal dust surface. (**a**) Schematic diagram of cationic surfactant adsorption on coal dust surface. (**b**) Schematic diagram of adsorption of anionic surfactant on coal dust surface.

**Figure 10 polymers-12-02942-f010:**
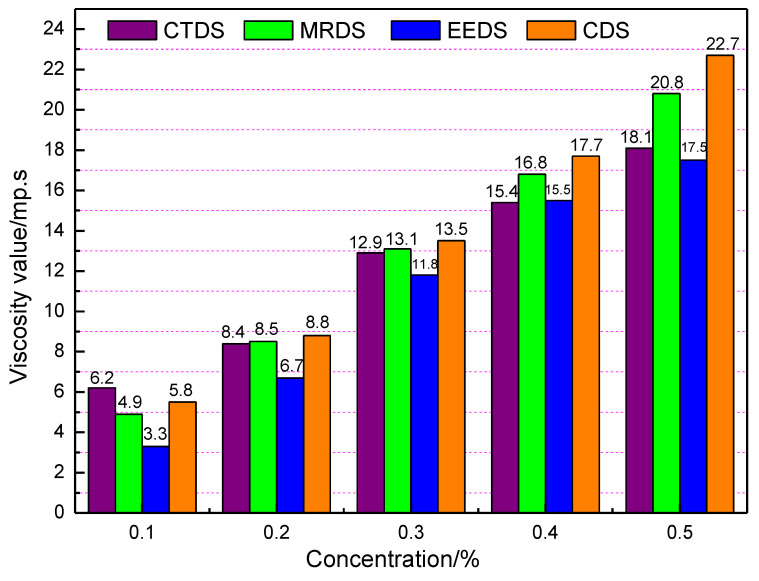
Histogram of viscosity values of four dust suppressor solutions at different concentrations.

**Figure 11 polymers-12-02942-f011:**
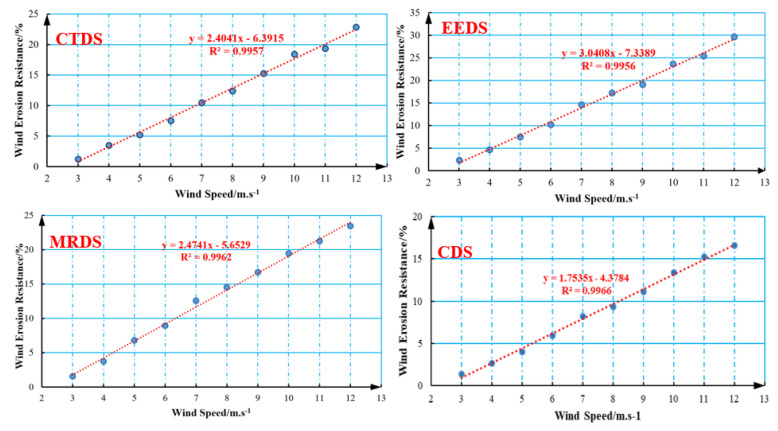
Linear fitting diagram of wind erosion resistance data of four dust suppressors.

**Figure 12 polymers-12-02942-f012:**
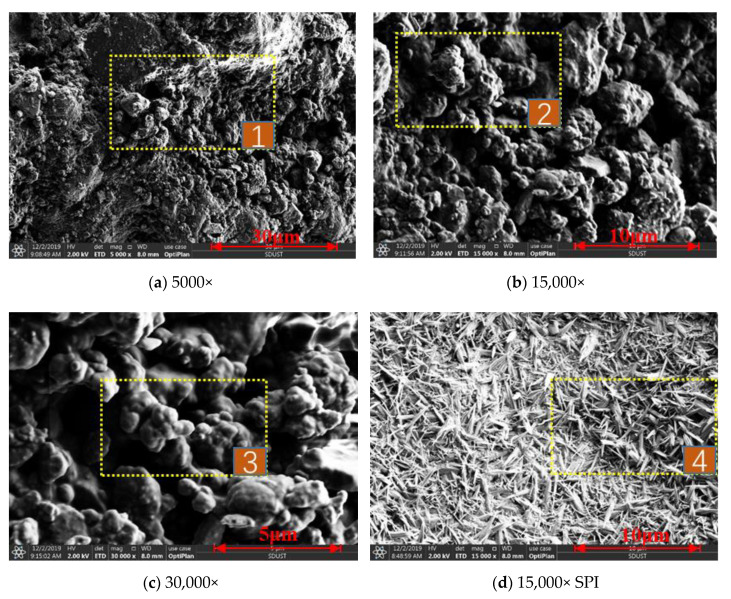
SEM images of dust suppressant and soy protein isolate. (**a**–**c**) are SEM images of 5000, 15,000, and 30,000 respectively enlarged by the dust suppressant sprayed on the surface of coal dust; (**d**) figure is an SEM image of dry CDS.

**Figure 13 polymers-12-02942-f013:**
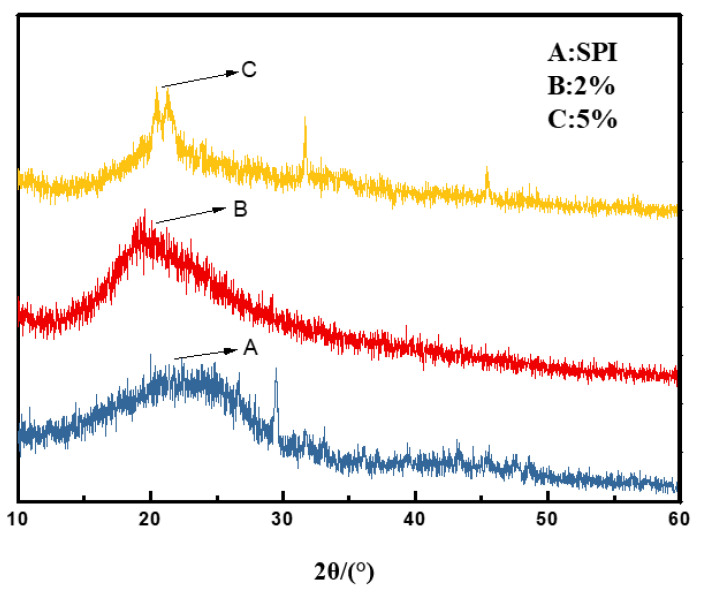
XRD diffraction pattern.

**Figure 14 polymers-12-02942-f014:**
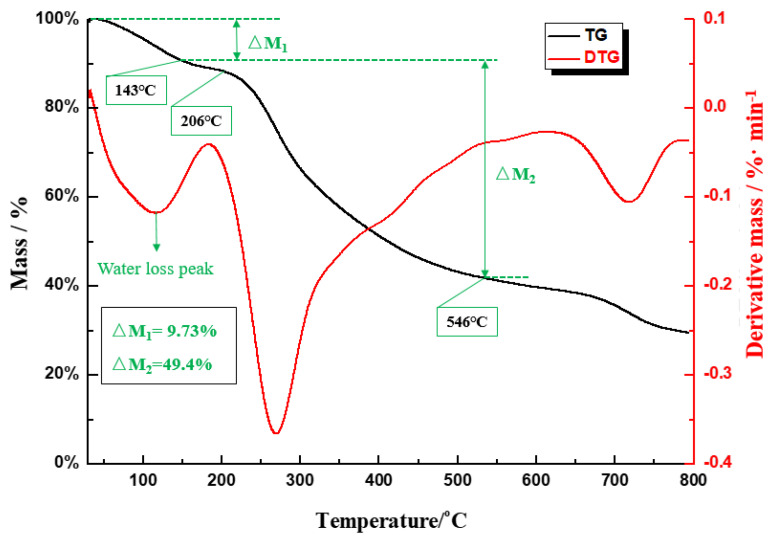
TG-DTG curve of dust suppressant sample.

**Table 1 polymers-12-02942-t001:** Main experimental reagents.

Types of Raw Materials	Raw Material Name	English Abbreviations	Purity	Manufacturer
**Main reagents**	Soy protein isolate	SPI	BR	Sinopharm Chemical Reagent Co., Ltd., China
	Sodium carboxymethyl cellulose	CMC	CP	Xiya Reagent Co., Ltd., China
	Sodium methyl silicate	SMS	GR	Shandong Yousuo Chemical Technology Co., Ltd., Shandong, China
**Surfactant**	Fatty alcohol polyoxyethylene ether	AEO	Tech	Shandong Yousuo Chemical Technology Co., Ltd., Shandong, China
	Cetyl Trimethyl Ammonium Bromide	CTAB	Tech	Shandong Yousuo Chemical Technology Co., Ltd., Shandong, China
	Sodium dodecyl sulfate	SDS	Tech	Shandong Yousuo Chemical Technology Co., Ltd., Shandong, China
**Dust Suppressant**	Crust-type dust suppressant	CTDS	Tech	Dacheng County Yibo Chemical Co., Ltd., China
	Mine road dust suppressant	MRDS	Tech	Langfang Tianshuo Chemical Technology Co., Ltd., China
	Efficient environmental dust suppressant	EEDS	Tech	Hebei Lankai Energy Saving Technology Co., Ltd., China

**Table 2 polymers-12-02942-t002:** Main experimental instruments and models.

Experimental Apparatus	Model Name	Manufacturer
**Malvern laser particle size analyzer**	Mastersizer 3000	Malvern, UK
**Dynamic contact angle measuring instrument**	KRUSS	KRüSS company
**Zeta Potentiometer**	ELSZ-2000	Suzhou Otsuka Electronics Co., Ltd, China
**Constant temperature magnetic heating stirrer**	85-1	Shanghai meiyingpu Instrument Manufacturing Co., Ltd., China
**Rotational viscometer**	NDI-79	Shanghai Precision Instrument Co., Ltd., Shanghai, China
**High resolution scanning electron microscope**	APREO	Shanghai Casting Gold Analytical Instruments and Equipment Co., Ltd., Shanghai, China
**X-ray diffractometer**	Rigaku Utima IV	Rigaku corperation
**Thermogravimetric Analyzer**	Labsys Evo	Mettler Toledo International Trading (Shanghai) Co., Ltd., China
**Electronic analytical balance**	ME104E	Tianjin Tianma Instruments Co., Ltd, China
**Vacuum drying oven**	DHG-9030	Shanghai Yiheng technology mailbox company, China

**Table 3 polymers-12-02942-t003:** Statistics of wind erosion rate of four dust suppressors.

Wind Speed		3 m/s	4 m/s	5 m/s	6 m/s	7 m/s	8 m/s	9 m/s	10 m/s	11 m/s	12 m/s
CTDS	M_1_	90.23	90.43	89.23	88.45	90.28	89.24	87.98	89.56	90.23	90.85
	M_2_	89.10	87.25	84.58	81.76	80.81	78.15	74.55	73.07	72.74	70.07
	δ_1_	1.25	3.52	5.21	7.56	10.49	12.43	15.27	18.41	19.38	22.87
MRDS	M_3_	88.35	89.56	88.65	89.23	89.28	89.67	88.79	88.63	89.25	88.39
	M_4_	86.96	86.22	82.60	81.26	78.07	76.65	73.94	71.37	70.28	67.65
	δ_2_	1.57	3.73	6.82	8.93	12.56	14.52	16.72	19.47	21.25	23.46
EEDS	M_5_	87.36	87.23	87.64	87.98	88.23	88.25	87.68	87.73	88.15	88.24
	M_6_	85.31	83.15	81.04	79.00	75.29	73.03	70.90	66.96	65.69	62.04
	δ_3_	2.35	4.68	7.53	10.21	14.67	17.25	19.14	23.67	25.48	29.69
CDS	M_7_	88.34	88.75	89.21	88.72	88.96	88.31	89.25	89.41	89.16	88.47
	M_8_	88.15	87.59	86.82	84.97	83.45	80.91	79.22	76.61	74.38	70.23
	δ_4_	0.22	1.31	2.68	4.23	6.19	8.38	11.24	14.32	16.58	20.62

Where δ is the wind erosion rate, wt%; Mn is the original mass (g); Mn + 1 is the residual mass after wind erosion (g).

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
