# Peer review of "Preparation and Characterization of a Composite Dust Suppressant for Coal Mines"

_polymers, 2020, doi:10.3390/polym12122942_

Round 1

Reviewer 1 Report

Comments on Manuscript polymer-993887

This manuscript reports the preparation of a composite dust suppressant, its properties and performance for coal powder.

Effective dust suppressants have important implications in the coal industry. It will benefit the readers in the industry and academia to share a detailed report on the preparation and characterization of efficient dust suppressant designed for coal. On the other hand, this paper is not well written, some statement/interpretation is incorrect. Therefore, I suggest that major revisions are needed before this report can be published.

Major concerns:

  1. The authors should revise the English very carefully.
  2. The structure of this paper is very confusing. The authors should follow the instruction/template of the journal to prepare the manuscript. It is especially important have a specific section of Materials and Method, and all the relevant information should not be scattered in the manuscript which makes it difficult to read.
  3. Figure 1, Figure 3 and Figure 13 are not needed.
  4. Details of the contact angle measurement are needed. How did the authors get the flat coal surface for CA measurement? Was there any polishing or other treating process? Please note the font of the words on Y-axis needs to be correct.
  5. Experimental details of the zeta potential test are needed: water composition? pH? Temperature? procedure to prepare coal particle dispersion? procedure of zeta potential measurement itself?
  6. Lines 167-170, the interpretation of the zeta potential change due to surfactant is not correct. the coal particles carry negative charge; nonionic surfactant does not change it. It does not mean it does not generate ion. Adding cationic surfactant will neutralize the negative charge first, then the overall charge will be positive.
  7. Line 180 to line 181, “anionic surfactant SDS ionized the negative charged surface active ions in the solution”. This sentence is difficult to understand.
  8. Figure 8, the bilayer structure needs to be corrected. It looks like a surfactant with two positively charged polar head, which is obviously wrong.
  9. Figure 9 and relevant discussion is wrong. Due to Coulombic interaction, it is unlikely the negatively charged head of surfactant can interact with the negatively charged surface. The anionic surfactant likely be adsorbed onto coal surface using the hydrophobic tail through van der Waal interaction.
  10. Lines 225-227, “when the concentration of CDDS exceeds 1%, it has good bonding properties and can more effectively inhibit airborne dust than the three commercially available dust suppressants.” This statement is wrong. High viscosity means the molecules in the solution have stronger interaction; but it does not mean they will have stronger interaction with the dust particles which are desired for better dust suppressant.
  11. Lines 306-308, “As shown in Fig. 15, the XRD patterns of the films of the soy protein isolate sample and CDDS at initial concentrations of both 2 and 5%, were basically the same, with both having a broad main peak at 2θ = 20°.” This statement is wrong. Curve A and curve B are obviously different.
  12. In Figure 16, please note that deltaH has very specific meaning in calorimetry. It does not stand for weight loss.

Minor concerns:

  1. A figure with the lab and the setup for the erosion rate testing presented in Table 5 are needed. Also, the detailed procedure should be included.
  2. Why did the authors use CDDS for “composite dust suppressant”? What does the second “D” stand for? It should be CDS.
  3. In the title, “suppressor” should be corrected to “suppressant”.
  4. Line 26, “suppression film” should be “suppressant film”.
  5. Please note “*” used by the authors in the unit of viscosity is incorrect.
  6. When the authors used “%” for percentage, they should clearly state it is by weight or by molarity.
  7. Line 43, please note the second “2019-nCoV” needs to be “2019-nCoV virus”.
  8. Please note the molecule for “SDS” in Figure 2 is not SDS, it is SDBS.
  9. The authors should give more details of the soy protein isolate in Table 1. What does BR mean?
  10. In Table 1, what is the molecular weight of the CMC? What about molecular weight of AEO?
  11. In Table 2, the authors need to give the brand and model name of the instrument, not just some confusing mixed information.
  12. Line 93, “Shandong Province” should be “Shandong Province, China.“
  13. Table 3 and Figure 4 overlap. Please just keep Figure 4.
  14. In Figure 4, what are the samples for the plots with different color?
  15. Line 199, “Fix the three-necked flask on a constant-temperature magnetic stirrer”. This sentence needs to be re-written in right English.

Author Response

To Reviewer #1:

We’d like to express our great appreciation to the reviewer for comments on our manuscript. The responses to the reviewer’s comments are presented as following:

Major concerns:

Comment 1:

The authors should revise the English very carefully.

Response:

Thanks for the valuable suggestion made by the reviewer. As per the reviewer's advice, we have sought assistance from professional English editors to help improve the grammar, spelling and sentence structures, as well as correcting errors and polishing the language. Please refer to the parts highlighted in red in the manuscript for the detail.

Comment 2:

The structure of this paper is very confusing. The authors should follow the instruction/template of the journal to prepare the manuscript. It is especially important have a specific section of Materials and Method, and all the relevant information should not be scattered in the manuscript which makes it difficult to read.

Response:

Thanks for the reviewer’s valuable suggestion. We have reorganized the structure of the manuscript. Please check.

Comment 3:

Figure 1, Figure 3 and Figure 13 are not needed.

Response:

Thanks for the valuable suggestion made by the reviewer. We have deleted Figure 3 and Figure 13. We want to keep Figure 1 in the manuscript. Figure 1 can more intuitively describe the harm caused by dust pollution. Miners working in a high-concentration dust environment for a long time can cause pneumoconiosis. At the same time, dust in the air as a virus carrier can spread the virus, which is also a hot research direction recently. Therefore, we want to keep Figure 1 so that readers can easily understand the significance of our research work. I implore the professor to understand our thoughts.

Comment 4:

Details of the contact angle measurement are needed. How did the authors get the flat coal surface for CA measurement? Was there any polishing or other treating process? Please note the font of the words on Y-axis needs to be correct.

Response:

Thanks for the reviewer’s helpful suggestions. We have supplemented the contact angle measurement procedure in the manuscript.

Firstly, 550 mg pulverized coal was pressurized to 500t / cm2 for 3min to make coal cake. The diameter of coal cake is 13mm, the thickness is 2 ~ 3mm, and the surface is smooth. Take the coal cake and put it on the slide of the sample table of the horizontal contact angle measuring instrument. Lift the sample table slowly. When the sample plate is about to touch the liquid drop, the operating instrument starts to take pictures. After the coal cake are pressed, the surface is smooth, without polishing treatment.We have modified the font of the words on Y-axis. Please check.

Comment 5:

Experimental details of the zeta potential test are needed: water composition? pH? Temperature? procedure to prepare coal particle dispersion? procedure of zeta potential measurement itself?

Response:

Thanks for the reviewer’s valuable suggestion. We have supplemented the experimental details of the zeta potential testing.

In this manuscript, the ELSZ-2000 zeta potential analyzer is used to measure the zeta potential of coal powder in different surfactant solutions.

(1) First, prepare surfactant solutions of different concentrations in deionized water. Each equal amount of coal powder is put into the surfactant solution for ultrasonic dispersion for 20 minutes, and the mixture is configured into a uniform suspension. The suspension was prepared in a neutral environment at a room temperature of 23°C.

(2) Pour the suspension into the sample cell and close the lid, and put the sample cell into the zeta potential measuring instrument.

(3) Open the test software, set the parameter values of the sample to be tested, such as shading, refractive index, concentration, dielectric constant, etc., and select three times of measurement.

(4) After waiting for the result to come out, record the data and save the result.

Comment 6:

Lines 167-170, the interpretation of the zeta potential change due to surfactant is not correct. the coal particles carry negative charge; nonionic surfactant does not change it. It does not mean it does not generate ion. Adding cationic surfactant will neutralize the negative charge first, then the overall charge will be positive.

Response:

Thanks for the reviewer’s helpful suggestions. We have revised the relevant statements in the manuscript, please check.

Comment 7:

Line 180 to line 181, “anionic surfactant SDS ionized the negative charged surface active ions in the solution”. This sentence is difficult to understand.

Response:

Thanks for the reviewer’s helpful suggestions. We have revised the relevant sentences in the manuscript.

Comment 8:

Figure 8, the bilayer structure needs to be corrected. It looks like a surfactant with two positively charged polar head, which is obviously wrong.

Response:

Thanks for the reviewer’s helpful suggestions. We have revised figure 8, please check.

Fig.9 Schematic diagram of adsorption of surfactant on coal dust surface.

(a) Schematic diagram of cationic surfactant adsorption on coal dust surface.

(b) Schematic diagram of adsorption of anionic surfactant on coal dust surface.

Comment 9:

Figure 9 and relevant discussion is wrong. Due to Coulombic interaction, it is unlikely the negatively charged head of surfactant can interact with the negatively charged surface. The anionic surfactant likely be adsorbed onto coal surface using the hydrophobic tail through van der Waal interaction.

Response:

Thanks for the reviewer’s helpful suggestions. We have revised figure 9. We very much agree with your opinion: “The anionic surfactant likely be adsorbed onto coal surface using the hydrophobic tail through van der Waal interaction.”

Comment 10:

Lines 225-227, “when the concentration of CDDS exceeds 1%, it has good bonding properties and can more effectively inhibit airborne dust than the three commercially available dust suppressants.” This statement is wrong. High viscosity means the molecules in the solution have stronger interaction; but it does not mean they will have stronger interaction with the dust particles which are desired for better dust suppressant.

Response:

Thanks for the reviewer’s helpful suggestions. We have revised the sentences in the manuscript. Please check it.

Compared with three commercially available dust suppressant solutions, the molecules in the solution have stronger interaction.

Comment 11:

Lines 306-308, “As shown in Fig. 15, the XRD patterns of the films of the soy protein isolate sample and CDDS at initial concentrations of both 2 and 5%, were basically the same, with both having a broad main peak at 2θ = 20°.” This statement is wrong. Curve A and curve B are obviously different.

Response:

Thanks for the reviewer’s valuable suggestion. We have revised the relevant statements in the manuscript. Please check it.

Comment 12:

In Figure 16, please note that deltaH has very specific meaning in calorimetry. It does not stand for weight loss.

Response:

Thanks for the reviewer’s valuable suggestion. We replace deltaH with deltaM in the Figure. Please check it.

Minor concerns:

Comment 13:

A figure with the lab and the setup for the erosion rate testing presented in Table 5 are needed. Also, the detailed procedure should be included.

Response:

Thanks for the reviewer’s valuable suggestion. We have put the pictures of the experimental test equipment and process in the corresponding position of the manuscript. At the same time, the test procedures are supplemented in detail.

Fig.4 Layout drawing of the erosion rate testing experiment equipment

Comment 14:

Why did the authors use CDDS for “composite dust suppressant”? What does the second “D” stand for? It should be CDS.

Response:

Thanks for the reviewer’s valuable suggestion. The second “D” has no meaning. In order to be consistent with the abbreviations of the other three dust suppressors, they are all four letters, so we have added one more “D”. But we think it's a mistake. All “CDDS” in our manuscript have been changed to “CDS”. Please check.

Comment 15:

In the title, “suppressor” should be corrected to “suppressant”.

Response:

Thanks for the reviewer’s careful reading. We have revised it in the manuscript. Please check.

Comment 16:

Response:

Line 26, “suppression film” should be “suppressant film”.

Thanks for the reviewer’s careful reading. We have revised it in the manuscript. Please check.

Comment 17:

Please note “*” used by the authors in the unit of viscosity is incorrect.

Response:

Thanks for the reviewer’s careful reading. We have revised it in the manuscript. Please check.

Comment 18:

When the authors used “%” for percentage, they should clearly state it is by weight or by molarity.

Response:

Thanks for the reviewer’s helpful suggestions. The percentages used in our manuscripts are all weight percentages. We will state it clear in our manuscript.

Comment 19:

Line 43, please note the second “2019-nCoV” needs to be “2019-nCoV virus”.

Response:

Thanks for the reviewer’s careful reading. We have revised it in the manuscript. Please check.

Comment 20:

Please note the molecule for “SDS” in Figure 2 is not SDS, it is SDBS.

Response:

Thanks for the reviewer’s careful reading. In Figure 2, we have changed the ball-and-stick of SDBS to that of SDS. Please check.

Comment 21:

The authors should give more details of the soy protein isolate in Table 1. What does BR mean?

Response:

Thanks for the reviewer’s valuable suggestion. Due to the design of the table, we did not elaborate on the details of soy protein isolate. I hope the professor can understand it. The soy protein isolate used in the experiment is a dispersed type. Soy protein isolate has gel properties and conjunctival properties. It is biodegradable, natural, and renewable.

BR is an abbreviation for Biological reagent. It represents the purity of soy protein isolate.

Comment 22:

In Table 1, what is the molecular weight of the CMC? What about molecular weight of AEO?

Response:

Thanks for the reviewer’s comment. The molecular weight of the CMC is 263.198. The molecular weight of the AEO is 89.88618.

Comment 23:

In Table 2, the authors need to give the brand and model name of the instrument, not just some confusing mixed information.

Response:

Thanks for the reviewer’s valuable suggestion. We have supplemented the relevant contents of Table 2, including the model name and manufacturer of the instrument.

Comment 24:

Line 93, “Shandong Province” should be “Shandong Province, China.“

Response:

Thanks for the reviewer’s careful reading. We have revised it in the manuscript. Please check.

Comment 25:

Table 3 and Figure 4 overlap. Please just keep Figure 4.

Response:

Thanks for the reviewer’s valuable suggestion. We have deleted table 3, please check.

Comment 26:

In Figure 4, what are the samples for the plots with different color?

Response:

Thanks for the reviewer’s careful reading. When we were conducting coal dust laser particle size testing, we tested the sample five times and obtained 5 curves with different colors. Putting these five curves in a graph can better grasp the particle size distribution of dust. This can avoid measurement errors.

Comment 27:

Line 199, “Fix the three-necked flask on a constant-temperature magnetic stirrer”. This sentence needs to be re-written in right English.

Response:

Thanks for the reviewer’s careful reading. We have corrected the sentences in the manuscript, please cheak.

Reviewer 2 Report

The paper “Preparation and characterization of a composite dust suppressor for coal mines” by Jin et al. investigates the preparation and characterization of a formulation with a practical application as dust suppressor.

The article does not follow the recommended structure therefore the authors must re-organize the entire manuscript. A special “experimental section-materials and methods” must be included.

  1. Figure 2 must be deleted.
  2. Figures 8 and 9 can be combined in one single figure. At this point, why there are 2 heads for the adsorbed surfactant schematization on the surface of coal?!
  3. In table 4 the values for the determination 1, 2 and 3 must be deleted and only the average value can be provided. Moreover, in Fig 11 are displayed the same values as in table 4 thus one of them (table 4 or figure 11) must be deleted.
  4. Lines 225-227: the authors must add some literature references for the discussion of the sentence regarding the viscosity.
  5. Figure 13 must be deleted
  6. Caption of figure 14 is not complete. Please add the information for a, b, c and d
  7. As a general remark, the authors have not discussed the obtained results in correlation with the literature data.   

In view of the above, I recommend the publication of this manuscript after major corrections.

Author Response

To Reviewer #2:

We’d like to express our great appreciation to the reviewer for comments on our manuscript. We have re-organized the structure of the manuscript, please check. The responses to the reviewer’s comments are presented as following:

Comment 1:

Figure 2 must be deleted.

Response:

Thanks for the reviewer’s valuable suggestion. Fig. 2 is a schematic diagram of the preparation of dust suppressor, which shows the reagents used in the synthesis process of dust suppressor concisely and vividly. At the same time, it can show the use of dust suppressor. For example, it is used for dust suppression in coal piling, coal transportation by railway and road. Readers can understand the content of the article directly. Therefore, we want to keep figure 2. Please understand our decision.

Comment 2:

Figures 8 and 9 can be combined in one single figure. At this point, why there are 2 heads for the adsorbed surfactant schematization on the surface of coal?!

Response:

Thanks for the reviewer’s helpful suggestions. We have combined Figures 8 and 9 in one single figure. The head of the surfactant adsorbed on the coal surface should be one. We have modified the figure.

Fig.9 Schematic diagram of adsorption of surfactant on coal dust surface.

(a) Schematic diagram of cationic surfactant adsorption on coal dust surface.

(b) Schematic diagram of adsorption of anionic surfactant on coal dust surface.

Comment 3:

In table 4 the values for the determination 1, 2 and 3 must be deleted and only the average value can be provided. Moreover, in Fig 11 are displayed the same values as in table 4 thus one of them (table 4 or figure 11) must be deleted.

Response:

Thanks for the reviewer’s helpful suggestions. We have deleted table 4, please check.

Comment 4:

Lines 225-227: the authors must add some literature references for the discussion of the sentence regarding the viscosity.

Response:

Thanks for the reviewer’s valuable suggestion. We have added three references closely related to viscosity. The reference numbers are 41, 42 and 43.

Comment 5:

Figure 13 must be deleted.

Response:

Thanks for the reviewer’s valuable suggestion. We have deleted Figure 13.

Comment 6:

Caption of figure 14 is not complete. Please add the information for a, b, c and d.

Response:

Thanks for the reviewer’s valuable suggestion. We have added a, b, c and d information to figure 14.

Comment 7:

As a general remark, the authors have not discussed the obtained results in correlation with the literature data.

Response:

Thanks for the reviewer’s helpful suggestions. We further discuss the obtained results of literature data in the manuscript.

Reviewer 3 Report

The paper is focused on the preparation and characterization of a composite dust suppressant. The topic falls within the scope of the journal. Presentation and discussion of the results should be revised. The organization of the paper should be improved. On this basis, I recommend its publication after the following revisions:

  • An additional paragraph (“Experimental section”) with all the experimental details for each technique should be reported.
  • The paragraph “Preparation of composite dust suppressant” should be included in the Experimental section
  • The presentation and discussion of zeta potential data should be revised. Specifically, the authors stated “In contrast, the anionic surfactant SDS ionized the negatively charged surface active ions in the solution. The negatively charged hydrophilic head was negatively charged, so the hydrophobic tail was adsorbed on the surface of coal dust particles through the interaction of the hydrophobic groups.” Which solvent (pure water, or electrolyte aqueous solution,…) was used for zeta potential experiments? Which charged surface active ions in the solution can interact with SDS? Being SDS anionic, repulsive intercations with the coal surfaces could be expected. Please revise these sentences by taking into account these considerations/questions.
  • The scale length within SEM images (Fig. 14) is not clear. Please check and revise.
  • Figure 16. The y-axes for TG and DTG curve should be “mass / %” and “derivative mass / % min-1”, respectively.
  • What is the process related to the mass loss at ca. 700 °C (TG curve of Figure 16)?
  • The authors investigated anionic, cationic and non ionic surfactant to control the hydrophilic/hydrophobic characteristics of coal dust. Similar studies were conducted on clay/surfactant composites. Similarly to the submitted MS, these papers (Current Opinion in Colloid & Interface Science, 35, 2018, 42; Journal of Colloid and Interface Science, 547, 2019, 361-369) evidenced that the charge of head group of the surfactants is effective to control the wettability and charge of clay particles. I suggest to report this consideration in the Introduction.

Author Response

To Reviewer #3:

We’d like to express our great appreciation to the reviewer for comments on our manuscript. The responses to the reviewer’s comments are presented as following:

Comment 1:

An additional paragraph (“Experimental section”) with all the experimental details for each technique should be reported.

Response:

Thanks for the reviewer’s helpful suggestions. We added details of all experiments to the manuscript.

Comment 2:

The paragraph “Preparation of composite dust suppressant” should be included in the Experimental section

Response:

Thanks for the reviewer’s helpful suggestions. We have adjusted the structure of the manuscript and added the preparation of dust suppressant to the experimental section.

Comment 3:

The presentation and discussion of zeta potential data should be revised. Specifically, the authors stated “In contrast, the anionic surfactant SDS ionized the negatively charged surface active ions in the solution. The negatively charged hydrophilic head was negatively charged, so the hydrophobic tail was adsorbed on the surface of coal dust particles through the interaction of the hydrophobic groups.” Which solvent (pure water, or electrolyte aqueous solution,…) was used for zeta potential experiments? Which charged surface active ions in the solution can interact with SDS? Being SDS anionic, repulsive intercations with the coal surfaces could be expected. Please revise these sentences by taking into account these considerations/questions.

Response:

Thanks for the reviewer’s valuable suggestion. We have revised the relevant statements in the manuscript, please check.

Comment 4:

The scale length within SEM images (Fig. 14) is not clear. Please check and revise.

Response:

Thanks for the reviewer’s helpful suggestions. We have modified the scale length of SEM images and replaced them in the manuscript. Please check.

Comment 5:

Figure 16. The y-axes for TG and DTG curve should be “mass / %” and “derivative mass / % min-1”, respectively.

Response:

Thanks for the reviewer’s valuable suggestion. We have replaced the names of the Y-axis of the TG and DTG curves, please check.

Comment 6:

What is the process related to the mass loss at ca. 700 °C (TG curve of Figure 16)?

Response:

Thanks for the reviewer’s careful reading. In the process of analyzing the thermogravimetric curve, we defined four stages, the fourth stage is carbonization stage, the temperature range is 546 - 800 ℃, including 700 ℃. Therefore, it belongs to carbonization stage at 700 ℃. Between 546 - 700 ℃, the rate of weight loss at this stage accounted for 5.72% of the total weight loss of pyrolysis.

Comment 7:

The authors investigated anionic, cationic and non ionic surfactant to control the hydrophilic/hydrophobic characteristics of coal dust. Similar studies were conducted on clay/surfactant composites. Similarly to the submitted MS, these papers (Current Opinion in Colloid & Interface Science, 35, 2018, 42; Journal of Colloid and Interface Science, 547, 2019, 361-369) evidenced that the charge of head group of the surfactants is effective to control the wettability and charge of clay particles. I suggest to report this consideration in the Introduction.

Response:

Thanks for the reviewer’s valuable suggestion. We have quoted these two valuable manuscript in the introduction. The references are numbered 39 and 40.

Round 2

Reviewer 1 Report

The authors have made significant revisions to the manuscript. All concerns for the previous version have been addressed accordingly. 

I suggest this revised version can be published.

Reviewer 2 Report

The paper can be published as it is. 

Reviewer 3 Report

The paper was correctly revised according to the reviewers' suggestions. I recommend its publication without any further changes.